# Polydopamine Ultrathin Film Growth on Mica via In-Situ Polymerization of Dopamine with Applications for Silver-Based Antimicrobial Coatings

**DOI:** 10.3390/ma14030671

**Published:** 2021-02-01

**Authors:** Zheng-Hao Huang, Shi-Wei Peng, Shu-Ling Hsieh, Rajendranath Kirankumar, Po-Feng Huang, Tsao-Ming Chang, Atul Kumar Dwivedi, Nan-Fu Chen, Hao-Ming Wu, Shuchen Hsieh

**Affiliations:** 1Department of Rheumatology Immunology, Kaohsiung Armed Forced General Hospital, 2 Zhongzheng 1st Rd., Kaohsiung 80284, Taiwan; renewerrenewer@hotmail.com (Z.-H.H.); td000125732@gmail.com (P.-F.H.); 2Institute of Medical Science and Technology, National Sun Yat-sen University, Kaohsiung 80424, Taiwan; chen06688@gmail.com; 3Department of Chemistry, National Sun Yat-Sen University, 70 Lien-Hai Rd., Kaohsiung 80424, Taiwan; peng168660@hotmail.com (S.-W.P.); r7.kirankumar@gmail.com (R.K.); atuldwivedi08@gmail.com (A.K.D.); 4Department of Seafood Science, National Kaohsiung University of Science and Technology, 142 Haijhuan Rd., Kaohsiung 81157, Taiwan; slhsieh@nkust.edu.tw; 5Division of Neurosurgery, Department of Surgery, Kaohsiung Armed Forces General Hospital, 2 Zhongzheng 1st Rd., Kaohsiung 80284, Taiwan; augustimin78@gmail.com; 6Division of Cardiology, Department of Internal Medicine, Kaohsiung Armed Forces General Hospital, 2 Zhongzheng 1st Rd., Kaohsiung 80284, Taiwan; motherbbs@gmail.com; 7School of Pharmacy, College of Pharmacy, Kaohsiung Medical University, 100 Shih-Chuan 1st Rd., Kaohsiung 80708, Taiwan; 8Regenerative Medicine and Cell Therapy Research Center, Kaohsiung Medical University, 100 Shih-Chuan 1st Rd., Kaohsiung 80708, Taiwan

**Keywords:** atomic force microscopy, autoxidation, molecular level, nanofilm, polydopamine

## Abstract

The development of polydopamine (PDA) coatings with a nanometer-scale thickness on surfaces is highly desirable for exploiting the novel features arising from the specific structure on the molecular level. Exploring the mechanisms of thin-film growth is helpful for attaining desirable control over the useful properties of materials. We present a systematic study demonstrating the growth of a PDA thin film on the surface of mica in consecutive short deposition time intervals. Film growth at each deposition time was monitored through instrumental techniques such as atomic force microscopy (AFM), water contact angle (WCA) analysis, and X-ray photoelectron spectroscopy (XPS). Film growth was initiated by adsorption of the PDA molecules on mica, with subsequent island-like aggregation, and finally, a complete molecular level PDA film was formed on the surface due to further molecular adsorption. A duration of 60−300 s was sufficient for complete formation of the PDA layer within the thickness range of 0.5−1.1 nm. An outstanding feature of PDA ultrathin films is their ability to act as a molecular adhesive, providing a foundation for constructing functional surfaces. We also explored antimicrobial applications by incorporating Ag nanoparticles into a PDA film. The Ag NPs/PDA film was formed on a surgical blade and then characterized and confirmed by SEM-EDS and XPS. The modified film inhibited bacterial growth by up to 42% on the blade after cutting through a pork meat sample.

## 1. Introduction

An innovative study in 2007 highlighted the propensity of polydopamine (PDA) for facile one-step coating of surfaces. This exploration generated tremendous attention among researchers and proved to be a significant platform for the modification of various material surfaces for useful applications [1,2,3]. The catechol-amine chemistry of dopamine (DA) can be exploited for the use of PDA as an adhesive layer on a variety of substrates including hydrophobic and hydrophilic substrates, and substrates of different chemical compositions [4,5,6,7,8,9]. Moreover, the actions of PDA as an adhesive interface for further functionalization or consequent coatings are quite fascinating. Since its innovation, PDA has drawn significant interest as a powerful coating/interfacial material for the modification of surfaces [4]. PDA-modified materials have been generated for potential applications in various fields such as antibacterial materials [10], biosensing devices [11], materials for controlling cell adhesion [12], bio-optoelectronic materials [13], membranes [14,15], and the energy, environmental, and biomedical fields [16]. PDA is synthesized from a commercially available DA precursor through auto-polymerization, where the process is quite easy and attractive [17]. The formation of a PDA layer on a substrate surface mainly occurs through the polymerization of DA. Therefore, examining the underlying mechanistic of the formation of PDA is an active research area. The mechanism behind the conversion of DA to PDA, including the intermediate chemical structures, is quite complicated. Therefore, insight into the formation of a PDA layer on a substrate with consideration of the structural signatures is important.

Bernsmann et al. investigated the growth of a PDA film on a silicon oxide substrate, where the film thickness of PDA was dependent on the reaction time. A minimum film thickness of 4 nm was documented in their report [18]. Another study presented the effect of different oxidants on the deposition kinetics and film thickness of PDA [19], with focus on obtaining a PDA film thickness of 45 nm in multiple immersion steps. Another report presented by Kim and coworkers describes enhanced oxidation of dopamine in an oxygenated environment to generate a smooth PDA film of approximately 4 nm thickness [20]. Zangmeister et al. used a short deposition time for PDA coating to obtain a film thickness of approximately 2.5−3 nm within 2 min and 4−10 nm within 5 min on a gold surface and presented chemical characterization of the PDA film [21]. Walker et al. presented the controlled coating of PDA on a silicon substrate with a film thickness of 13.2 nm. Their work was mainly focused on enabling protein resistance [22]. Lamprou et al. studied the adhesiveness of PDA with a film thickness of 50 nm on polymer films [23]. Marco d’Ischia studied the growth of PDA films by varying the concentrations and interactions with different molecules. A minimum and maximum film thickness of around 5 nm and 60 nm was respectively obtained in their study [24]. Zeng and Xu et al. studied the deposition and adhesion of PDA on a gold surface as a function of the wettability [25].

All these reports demonstrate PDA film growth on substrate surfaces with higher film thicknesses. This itself implies the existence of multi-layered PDA films in many reports, where the films consist of aggregated PDA molecules on the surfaces. From the molecular point of view, the maximum molecular length of a single DA molecule in free space, calculated by using the Cambridge Crystallographic Data Centre software (Figure A2) [26], is around 0.78 nm. If we compare the maximum molecular length of a single DA molecule with the reported film thicknesses, we see that polymerization and concomitant aggregation of the PDA molecules would generally generate higher order aggregates through π-π interaction on the substrate surfaces, thus increasing the PDA film thickness to several nanometers rather than the unit value, where the former is generally seen in the literature reports. A significant report supporting our hypothesis was presented recently by Buehler and coworkers [27]. This report provided detailed elucidation of the structural features of the dimer, trimer, tetramer, and even higher-order PDA oligomers, upon polymerization on the basis of a computational method. The propensity of PDA molecules to form higher-order aggregates was well outlined in their report.

In contrast with these reports, our efforts are focused on developing a molecular level thin film of PDA on a mica surface. It is well accepted that thin films of molecules on a substrate surface are highly important due to their novel features arising at the structural level, where the features are directly linked to the molecular basis of the materials. It is thus compulsory to explore the mechanisms controlling thin film growth of materials on substrate surfaces, especially to achieve the desired control over novel properties of materials. In this context, utilization of an atomically smooth surface is well needed and suitable for exploring the short-range interactions that occur within a separation of less than 3 nm [28]. Our aim is thus to examine the growth of a PDA thin film in the minimum deposition time necessary for forming a continuous film on a mica surface and simultaneously to establish the physicochemical signatures of the PDA films. We thus systematically monitored the growth of a PDA layer on a mica surface in multiple steps over short deposition time intervals. Growth of the PDA film was monitored through atomic force microscopy (AFM) and water contact angle (WCA) analysis, and further structural features were deduced from X-ray photoelectron spectroscopy (XPS) under various deposition times.

We demonstrate a systematic study of PDA film growth within the thickness range of 0.5−1.1 nm in specific and short deposition time intervals from 60 to 540 s. The film thickness of PDA ranging from 0.5−1.1 nm presented herein is rare, and studies focusing on the growth of PDA nanofilms on an atomically smooth mica surface, specifically in a step-wise manner, are even less explored. In the literature, efforts have been mainly devoted to exploring the effect of parameters such as the dopamine concentration, pH, and different oxidants and molecules on the thickness of PDA films. Therefore, our study specifically provides insight into the step-wise growth of a PDA film on the atomically smooth surface.

By understanding the molecular level property of the PDA, we demonstrate its application in the antibacterial property by coupling with Ag NPs. Ag NPs are strong antimicrobial and antibacterial agents, exhibiting high selectivity toward bacteria and low cytotoxicity toward human cells [29,30]. Because of this, Ag NPs are used in biomedical applications, dental components, surgical meshes, artificial joint replacements, and in various therapeutical applications. The PDA acts as an adhesive to incorporate Ag NPs and medium for interaction with the bacterial membrane. The mechanism for the antibacterial activity results from the release of Ag^+^ ions from the Ag NPs, which tend to bind with the surface of the bacterial membrane by interacting with sulfur and phosphorous-containing groups in the cell and cytoplasm, controlling cell division and inhibiting the growth of the bacterial cell [31,32].

## 2. Materials and Methods

### 2.1. Deposition of Polydopamine (PDA) Nanofilm on Mica

Dopamine hydrochloride (2 mg/mL; Sigma, St. Louis, MO, USA) in 10 mM tris(hydroxymethyl) aminomethane buffer solution (DP/Tris) at pH = 8.5 was prepared as reported in the previous literature [5,20,33]. Cleaved mica (Nilaco, Chuo city, Tokyo, Japan) was vertically dipped in DP/Tris for different immersion times (60−540 s). The obtained samples were then rinsed with purified water (Milli Q, 18.2 MΩ-cm at 25 °C), followed by drying under nitrogen flow at ambient temperature to obtain PDA nanofilm deposited mica (PDA/mica).

### 2.2. Deposition of Ag NPs/Polydopamine Nanofilm on a Surgical Blade

PDA nanofilm was deposited on the surgical blade using a similar procedure as described earlier. The modified surgical blade coated with PDA was further deposited with 0.3 M AgNO_3_ solution for 3 h at room temperature. The Ag NPs modified PDA on the surgical blade was rinsed with purified water (Milli Q, 18.2 MΩ-cm at 25 °C), followed by drying under nitrogen flow at ambient temperature.

### 2.3. Bacterial Growth Activity

The Ag NPs/PDA modified surgical blade was then used to cut on the pork sample (local meat market) and put the blade sample in the culture solution (Luria-Bertani (LB) medium). In the shaker, the bacteria are cultured in a 37 °C incubator and shaking for 2, 4, 6, and 8 h, and the BioTek Eon microplate spectrophotometer (BioTek Instruments, Winooski, VT, USA) was used to measure the antibacterial rate.

### 2.4. Characterization

The cleaved mica and prepared PDA/mica were characterized by atomic force microscopy (AFM, MFP-3D, Asylum Research, CA, USA) under ambient conditions. An AFM tip (AC240, Olympus, Tokyo, Japan) with a spring constant of 2.0 N m^−1^ was used, and the AFM was operated in AC mode with a resolution of 512 × 512 pixels. In tapping mode, the cantilever is oscillated at (or near) its resonance frequency and raster-scanned over an area of the surface while maintaining a feedback-controlled constant amplitude. The surface topography is derived from the cantilever “height” adjustments required to keep the amplitude constant. Roughness values (Rq) were determined from the topography image “height” data using the AFM system built-in analysis software.

The water contact angles of the samples were measured by using a PGX model instrument (Deerlijk, Belgium) under ambient conditions. Approximately 3.0 µL of purified water (Mili-Q, 18.2 MΩ-cm at 25 °C) was dropped on each sample surface. The reported contact angle is the average from five measurements. The surface energy was calculated by following the ASTM D5946 standard. A section of the PDA thin film on mica is inserted into the PGX model instrument. The syringe on top applies a single drop of deionized water onto the test specimen. An enlarged projection of the water drop is displayed on a CCD at the end of the PGX model instrument. The gauge is then adjusted to measure the angle between the material and the edge of the drop. Indirect measurement of the static contact angle can be made by calculation based upon measurements of the droplet silhouette according to the following Equation (1): Θ is the contact angle, H is the drop height image, and R is the half width.
Θ = 2 × arctan H/R(1)

The chemical composition of the prepared samples was analyzed by X-ray photoelectron spectroscopy (XPS) using a PHI-5000 VersaProbe instrument with a monochromated Al-Kα X-ray source, at an angle of 45°. The maximum length of DA and the unit cell lengths of four DA molecules were calculated by using the Cambridge Crystallographic Data software that is available free of charge via https://www.ccdc.cam.ac.uk/structures/search?id=doi:10.5517/cc10m9nl&sid=Data Cite.

## 3. Results

Despite the variety of existing coating methods, PDA thin films can be easily deposited by simply immersing a substrate surface into an aqueous alkaline solution of dopamine (DA). PDA formation is believed to occur by oxidative polymerization of DA under alkaline conditions. Such a simple dip-coating method has been vastly utilized by several research groups for coating various inorganic and organic surfaces. Moreover, the advantageous catechol-amine chemistry induces varied chemical interactions, resulting in enhanced wet adhesion and cohesion on a variety of materials. Based on this concept and the simple coating protocol, we first prepared dopamine solution in Tris buffer at pH 8.5 and then immersed the freshly cleaved mica in this solution for different time periods. The alkaline conditions led to the conversion of DA to PDA, with concomitant generation of a conformal PDA film on the surface of mica. The PDA films formed on the mica surface with different immersion times were then subjected to further characterization to deduce the structural signatures of PDA layer formation on the mica surface.

### 3.1. AFM Topography of PDA Film

The morphological signatures of the PDA thin film on mica were determined through atomic force microscopy (AFM). AFM topographical images and respective line-scans for the cleaved mica in the absence and presence of the PDA coatings formed by immersion for different times (60, 90, 120, 150, 180, 210, 300, 420 and 540 s) over a specific 1 × 1 μm^2^ scan area were recorded. The surface of the cleaved mica was smooth enough, with a measured root-mean-square (RMS) roughness of 21.4 pm, as shown in Figure 1a. Upon dipping the mica substrate into the DP/Tris solution, the surface topography of changed completely (Figure 1b−j). The modified surface topography of mica can be well attributed to the polymerization of DA molecules via polymer nucleation process and growth in solution with concomitant molecular adsorption on the mica surface. The line-scans for the samples immersed for up to 90 s indicated that the height of the PDA particles reached up to about 0.5 nm (Figure 1b,c), which is basically comparable to the horizontal length (0.25 nm) and vertical length (0.8 nm) of a single dopamine molecule [33]. This indicates that the PDA molecules/particles were dispersed as a monolayer on the mica surface. When the immersion time was further increased from 90 to 210 s (Figure 1c−g), the molecular particles of PDA became aggregated and gradually formed larger island-like structures with an approximate height of 1.1 nm on mica. During the process where the height increased, the relative RMS values for the immersion times of 120, 150, 180, 210, 300, 420, and 540 s varied from 158.6 to 266.6 pm. Therefore, an increased surface roughness upon PDA layer formation has been visualized. In 300 s of immersion time, RMS value reached to 242.1 pm with a film thickness of 0.5 nm. It is postulated that at this point, a complete PDA nanofilm layer was formed on mica. The formation of the complete PDA layer on the mica surface could be described by the Volmer–Weber (VW) growth mode [34], where island growth of PDA aggregates occurred via strong interactions such as π-π interaction and hydrogen bonding [35] between the PDA molecules. Such events should be responsible for the growth of the complete PDA layer. Notably, the RMS and height of the PDA particles decreased significantly for higher immersion times (from 300 to 540 s). Therefore, longer immersion times produced a less dramatic change in the height and roughness of the PDA layer. Immersion times longer than 300 s led to topographical images (Figure 1i,j) consisting of more dark features, indicative of significant gaps in the deposited PDA nanofilm. The corresponding line-scans in Figure 1i,j demonstrate that the height of the PDA molecules/particles reached up to 1.1 nm, possibly due to the formation of a second layer of PDA. Therefore, 300 s was sufficient for achieving a complete PDA layer (nanofilm) on the mica surface. Furthermore, the maximum molecular length of single DA (0.78 nm) and the unit cell lengths of four DA molecules (0.58, 0.88, and 1.46 nm) calculated through the Cambridge Crystallographic Data software were within the range of 0.5–1.5 nm. Therefore, this range further supports the existence of a molecular level PDA nanofilm on the PDA film.

### 3.2. Surface Coverage

The surface coverage ratio was determined by using particle analysis software (Igor Pro 6.36, MFP3D). The projection area of the particles in a 1 × 1 μm^2^ area range was calculated. The coverage ratio (%) was calculated by using the following equation:(2)$$\mathrm{Coverage}\;\mathrm{ratio}\;\left(\%\right)=\;\frac{\mathrm{Projection}\;\mathrm{area}\;\mathrm{of}\;\mathrm{particles}\;\left({\mu m}^{2}\right)}{1\times 1{\;\mu m}^{2}\;\mathrm{total}\;\mathrm{scan}\;\mathrm{area}}\;\times 100$$

Figure 2 shows the coverage ratio at different immersion times. There are two linear stages including (1) the steep increase of coverage ratio within less than 150 s of immersion time and (2) the slight increase of coverage ratio at longer immersion times. The steep curve appeared because of rapid adsorption of the PDA molecules on the empty space of the cleaved mica surface. The region of slight increase is possibly due to the large diffusion resistance due to the self-aggregation of PDA particles and the small gaps in the coating.

Thus, the mechanism of PDA deposition on the cleaved mica can be proposed as illustrated in Figure 3. Initial immersion of the mica surface into fresh alkaline DA solution led to the diffusion of DA molecules on the cleaved mica surface (Figure 3b). Diffusion of the DA molecules on the surface might be attributed to the interaction capability of the anchoring groups. In this stage, DA molecules would be present on the surface as well as in the solution. Afterward, simultaneous initiation of autoxidation and further polymerization of DA molecules would result in the presence of PDA on the surface and in solution. The generated PDA molecules would thus further act as seeding materials for enhancing the overall autoxidation of the DA molecules. Therefore, DA would be converted to PDA more rapidly, with simultaneous growth over time. Because PDA molecules have a large number of anchoring groups with strong interactions compared to their monomer counterpart, a simultaneous increase in the surface coverage of the PDA molecules was observed. At longer immersion times, the PDA molecules/particles aggregates became larger and deposited slowly into the gaps or on the deposited PDA molecules/particles (Figure 3c) [8,36], and finally deposition of molecules was completed to generate a complete PDA layer (Figure 3d). The adsorption and coating of PDA on mica, explained by the Volmer–Weber growth model, can be further summarized in a stepwise manner as (1) PDA molecules were adsorbed on the mica surface and served as the seeding materials, (2) other PDA particles were adsorbed around the seeding materials to form island-like PDA particles, and (3) other available PDA particles were deposited in the gaps between the island-like particles or on the surface of the native particles.

### 3.3. Water Contact Angle

To obtain further insight into the formation of the PDA film on the mica surface, we further investigated the nature of the coated surface. Evaluation of the water contact (WCA) is one of the most common methods of measuring the surface wettability. Therefore, a wettability test was performed to explore the hydrophobicity and/or hydrophilicity of the PDA-coated mica surface. The WCA and surface energy were plotted as a function of the immersion time (0 to 540 s), as presented in Figure 4. For immersion times shorter than 300 s, the WCA increased dramatically, with concomitant and rapid reduction of the surface energy. However, immersion times greater than 300 s led to less dramatic variations in the WCA and surface energy. Only slight changes were observed when the immersion time was longer than 300 s. Such variations adequately explain the significant changes in the hydrophobicity of the PDA-coated mica surface. Specifically, the hydrophobicity of the mica surface was significantly enhanced. Such a significant increment in the hydrophobicity of the mica surface may be attributed to the structural signatures of the PDA molecules, where π-π interactions and hydrogen bonding interactions among molecules dominantly occurred during film formation. Upon reaching a certain molecular population threshold on the mica surface, the hydrophobicity could enhance slightly. For these reasons, a slight increment in the WCA and slight decrement in the surface energy was observed for immersion times longer than 300 s. Notably, coverage of the mica surface by PDA molecules was achieved within 300 s, which further validates the morphological signatures obtained by AFM.

Table 1 shows RMS roughness values from AFM images compared with contact an-gle measurements obtained at different times from 0 to 540 s.

### 3.4. XPS Analysis

The formation of a PDA film on a substrate surface is accompanied by polymerization of dopamine. However, it is still hard to establish the exact structural signatures of PDA and the intermediates. Several factors influence the formation of the PDA layer on a substrate, and the structural features of such layers are unknown. Therefore, efforts toward elucidating these factors and features are important. The structural features of the PDA layer on the mica substrate were evaluated by using XPS spectroscopy. XPS spectral features may provide valuable information about the chemical composition and chemical environments of PDA on the mica surface. Figure 5 shows the C 1S and N 1S XPS spectra of the samples prepared with different immersion times (0, 60, 150, 300, and 540 s). The C 1S spectra were curve fitted into three peak components corresponding to C–H and C–NH_2_ at 284.6 eV, C–OH and C–N at 285.8 eV, and C = O at 288.3 eV [21,37]. The N 1S spectrum was curve fitted into three peak components of a tertiary amine (R–N=) at 398.6 eV, secondary amine (R_2_–NH) at 399.9 eV, and primary amine (R–NH_2_) at 401.7 eV [21,38]. When the immersion time was increased, the spectral peaks related to C 1S and N 1S were dominated by C–H and C–NH_2_ functionalities centered at 284.6 eV and 399.9 eV (R_2_–NH), respectively. The nitrogen to carbon ratio (N/C) calculated from the ratio of N 1S and C 1S peak area was found in the range between 0.084 and 0.112. The calculated N/C ratios are close to the reported theoretical values of dopamine (N/C = 0.125) [5]. This result confirmed that the polymerization of dopamine and film formation occurred simultaneously. Henceforth, the stages of dopamine polymerization can be monitored by the XPS spectrum.

Table 2 shows the peak integral values of different functional groups under different immersion times. The increased intensity of C=O was attributed to the formation of intermediate species and tautomers (Figure A1). The C 1S region includes more than one component at the same binding energy (285.8 eV), including C–OH and C–N functionalities. Therefore, the discussion presented herein mainly focuses on the N 1S peak intensities. A close look at the chemical structure of PDA [13,39] reveals that the R–NH_2_ group belongs to dopamine, while R_2_–NH is found in both the intermediate and PDA. The R–N= functional group corresponds to the dopaminechrome and tautomeric species. These intermediate species might be 5,6-dihydroxyindole and 5,6-indolequinone. The peak intensity of R–NH_2_ decreased with increasing immersion time. This is because of the conversion of dopamine into reaction intermediates and tautomeric species via oxidation and cyclization. Specifically, increasing the immersion time increased the population of R–N= species relative to that of dopaminechrome formed through the oxidation of leuco-dopaminechrome; the tautomeric species were 5,6-dihydroxyindole and 5,6-indolequinone (Figure A1). Furthermore, the components with R_2_–NH groups grew to the highest value in N 1S region under immersion time longer than 150 s. This could be related to the formation of PDA. Thus, the PDA film contained not only PDA, but intermediates and tautomeric species, as well as dopamine.

The formation mechanism for PDA formation and intermediate species (5,6-dihydroxyindole and 5,6-indolequinone) are shown in Figure A1 [7,17,21,22,23,24,25,26,27]. The intermediate 5,6-indolequinone results from the oxidation of phenolic hydroxyls, and cyclization of the pendant amine occurs simultaneously [4]. However, efforts in this direction must be more precise to elucidate the PDA structure.

### 3.5. Antibacterial Properties of Ag/PDA Composite

Dopamine was auto-polymerized on the surgical blade using a similar technique reported earlier to modify on mica Figure A3. In which Ag nanoparticles were deposited on surgical blade coated with PDA following our previously reported conditions with AgNO_3_ solution [40]. We have used XPS and SEM-EDX to characterize Ag in the film coatings as shown below in Figure A4 and Figure A5. The XPS data present high-resolution spectra of the Ag NPs/polydopamine nanofilm on a surgical blade (Figure A4). As shown in Figure A4, the doublet peaks at 378 and 372 eV, corresponding to Ag 3d3/2 and Ag 3d5/2 of elemental silver, confirm the successful reduction of Ag nanoparticles on the polydopamine layer. The corresponding SEM images were taken without and with the modification of Ag NPs/PDA on a surgical blade (Figure A5a,e). Elemental mapping was performed to identify Ag as shown in Figure A5b,f. In Figure A5c,g the mapping image of Ag was superimposed on the SEM image. The corresponding EDS spectra were obtained for a surgical blade as shown in Figure A5d and Ag modified on PDA in Figure A5h. The presence of Ag in EDS confirms the successful modification of Ag/PDA on the surgical blade. A clean surgical blade control sample was imaged with SEM before and after cutting a pork meat sample, as shown in Figure 6a,b, respectively. Meat residue on the blade was clearly observed in Figure 6b, which may act as a medium for growing bacteria. SEM images of a modified Ag NPs/PDA surgical blade before and after cutting a pork meat sample are shown in Figure 6c,d, respectively. Comparing Figure 6a,d, it is apparent that the Ag NPs/PDA film is evenly distributed on the surface. After cutting the pork meat sample, traces of meat residue are clearly present on the Ag NPs/PDA modified surface (Figure 6d). This sample was subjected to further analysis. In Figure A6, the surgical blade control, with PDA and Ag/PDA composite were used to cut pork meat samples. The respective blade was directly tested for antibacterial activity with respect to time. The growth rate and bacterial concentrations were determined by measuring optical density (O.D. at 570 nm each hour). The result shows that antibacterial activity of the Ag NPs/PDA is much higher than that of PDA alone. This can be seen clearly with the plot of O.D. vs. time in Figure A6d. The growth of *E.coli* bacteria on the blade was monitored over a period of 8 h using absorbance at 570 nm. PDA alone does not show any significant antibacterial inhibition property when coupled with Ag NPs growth curve of bacteria remains less showing a superior antibacterial activity. The growth inhibition curve was found to be 42% less when compared with the control sample. In our previous studies we have shown that the antibacterial properties of Ag NPs are due to the release of silver ions (Ag+), which interact with the thiol groups present in bacteria forming S-Ag or disulfide bonds. This interaction affects bacterial proteins and interferes with the DNA replication process. Thus, exposure to Ag NPs attached to PDA causes the bacterial cell membrane to collapse resulting in cell death and the release of intercellular components.

## 4. Conclusions

In conclusion, we systematically monitored the growth of a PDA film on a mica surface in short deposition time intervals by monitoring the AFM topography, roughness parameters, and WCA. AFM measurements revealed that a 0.5 nm thin film of PDA was formed in 90 s that was comparable to the molecular length of a single DA molecule [33]. Subsequent immersion times ranging from 90 to 210 s led to an increase in the film thickness up to 1.1 nm on mica. Furthermore, the maximum molecular length of single DA (0.78 nm) and the unit cell lengths of four DA molecules (0.58, 0.88, and 1.46 nm) obtained from the Cambridge Crystallographic Data software [26] were within the comparable range of 0.5 to 1.5 nm. The common parameters, such as the surface roughness, surface coverage ratio, and WCA variations were highly significant within 0 to 300 s. However, immersion times greater than 300 s resulted in less dramatic changes and had the least impact on the film thickness and variation of other parameters. Therefore, our efforts are much more focused on the development of a molecular level thin film of PDA on a surface rather than multilayered films. Surface modification of materials to adjust the desired physicochemical properties of materials is an important research topic for the construction of composite materials with combined properties. This study provides significant insight into the formation of PDA nanofilms (molecular level) on a surface. We believe that the performance and outcomes of PDA-based materials and interfaces can be further improved by utilizing conformal PDA coatings within the nanometer-scale. Understanding the molecular layer property of PDA helps in the layer-by-layer assembly of Ag NPs on it and to show application towards the antibacterial property.

## Figures and Tables

**Figure 1 materials-14-00671-f001:**
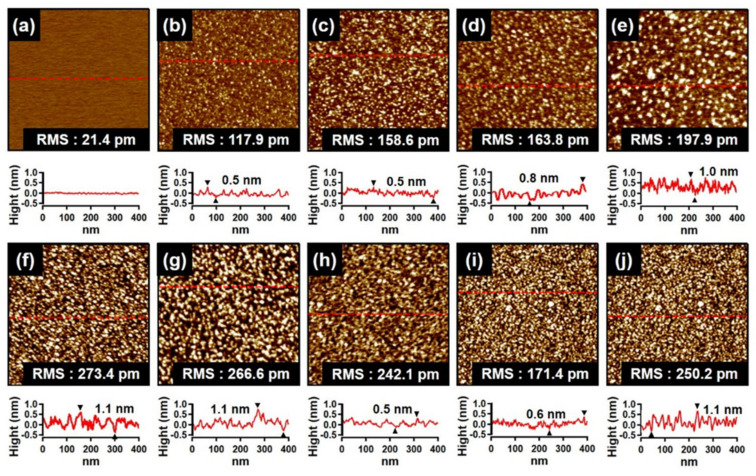
AFM topography images and line−scans of (**a**) cleaved mica and PDA/mica prepared by immersion for different times: (**b**) 60, (**c**) 90, (**d**) 120, (**e**) 150, (**f**) 180, (**g**) 210, (**h**) 300, (**i**) 420, and (**j**) 540 s, respectively. The line−scans are presented below each topography image.

**Figure 2 materials-14-00671-f002:**
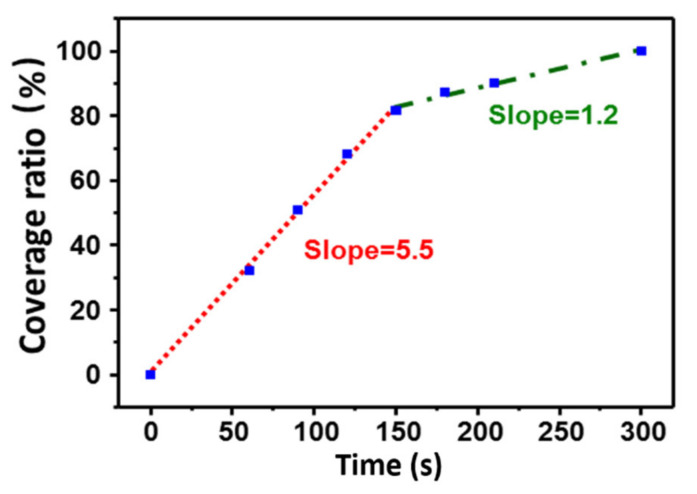
Coverage ratio (%) as a function of the immersion time for formation of complete PDA layer.

**Figure 3 materials-14-00671-f003:**
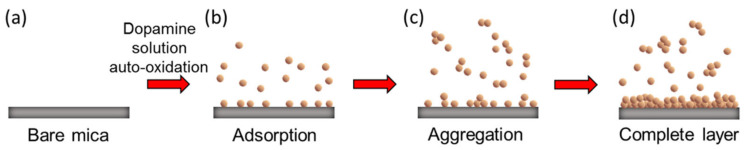
Schematic of mechanism of deposition of PDA on mica surface (**a**) bare mica, (**b**) adsorption of dopamine on mica surface, (**c**) aggregation of dopamine, and (**d**) layer formation on mica surface.

**Figure 4 materials-14-00671-f004:**
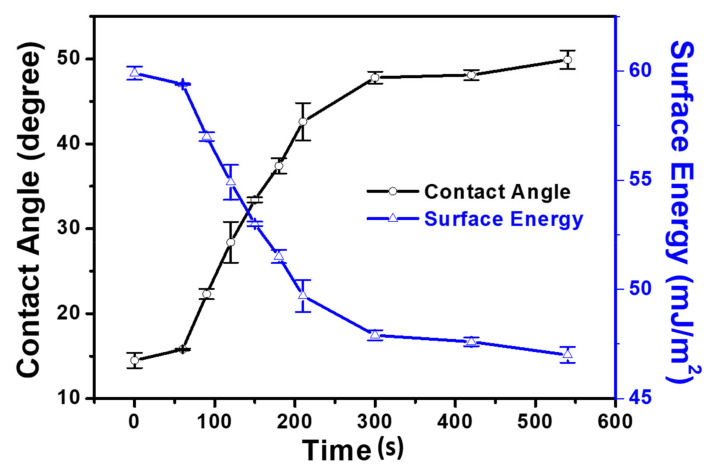
Plot of water contact angle (degree) and surface energy (mJ/m^2^) versus time for PDA/mica.

**Figure 5 materials-14-00671-f005:**
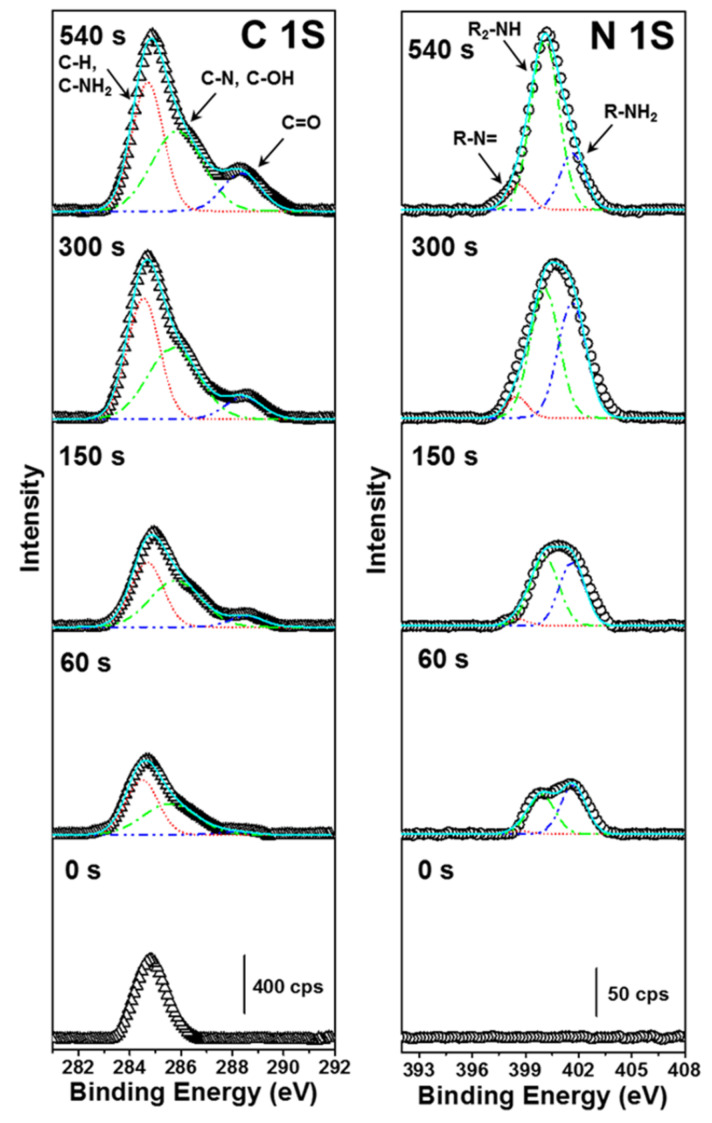
XPS C 1S and N 1S spectra of polydopamine deposited on mica for different immersion times (0, 60, 150, 300, and 540 s).

**Figure 6 materials-14-00671-f006:**
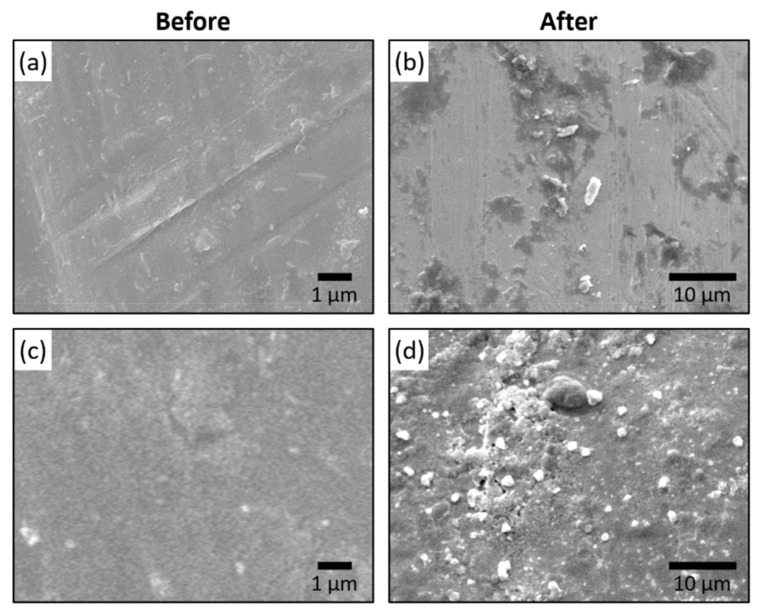
SEM image of surgical blade (**a**) before and (**b**) after cutting the meat sample. Ag/PDA modified surgical blade (**c**) before and (**d**) after cutting the meat sample.

**Table 1 materials-14-00671-t001:** RMS and contact angle data of the cleaved mica and PDA/mica prepared by immersion for different times (60, 90, 120, 150, 180, 210, 300, 420, and 540 s), respectively.

Time (s)	0	60	90	120	150
RMS (nm)	21.4 ^a^	117.9 ^b^	158.6 ^c^	163.8 ^c,d,i^	197.9 ^e^
contact angle (°)	14.5 ± 0.9 ^a^	15.8 ± 0.1 ^a^	22.3 ± 0.6 ^b^	28.4 ± 2.4 ^c^	33.4 ± 0.3 ^d^
**Time (s)**	**180**	**210**	**300**	**420**	**540**
RMS (nm)	273.4 ^f^	266.6 ^d,f,g^	242.1 ^h^	171.4 ^i^	250.2 ^g,h^
contact angle (°)	37.4 ± 0.9 ^e^	42.6 ± 2.2 ^f^	47.8 ± 0.7 ^g^	48.1 ± 0.6 ^g,h^	49.9 ± 1.1 ^h^

All data are mean ± standard error of mean (n = 3); a–i, letters indicate data that do not differ at significance level *p* < 0.05.

**Table 2 materials-14-00671-t002:** Chemical composition of PDA/mica derived from C 1S and N 1S signals based on immersion time.

Time (s)	Composition (%)					
	CH and C–NH_2_	C–OH and C–N	C=O	=N–R	R_2_–NH	R–NH_2_
60	50.0	46.9	3.1	3.0	44.6	52.4
150	43.1	49.3	7.6	3.7	52.2	44.1
300	46.5	43.8	9.7	6.1	50.1	43.8
540	42.1	42.7	15.2	8.5	70.5	21.0

## Data Availability

Data sharing is not applicable to this article.

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
