# Peer review of "Polydopamine Ultrathin Film Growth on Mica via In-Situ Polymerization of Dopamine with Applications for Silver-Based Antimicrobial Coatings"

_materials, 2021, doi:10.3390/ma14030671_

Round 1

Reviewer 1 Report

The author has shown the molecular level polydopamine thin film growth on mica surface via in-situ polymerization of dopamine. The framework and overall design for the experiment conducted are well but required major revisions for further consideration. I, therefore, recommend the paper should be revised and resubmitted for consideration.

  1. Mostly this work is depending on Ag and its role, but not included in the title, so the title should be revised
  2. The abstract should be improved with the detailed results achieved. The application of this study is also missing in the abstract.
  3. The Introduction is a lack of aim and evidence of similar recent work, and it should be revised with recent research works.
  4. The importance of Ag is missing in the introduction.
  5. Only AFM imaging is not enough to confirm the Ag presence, hence, XRD, SEM-EDX imaging with elemental mapping is required to prove the coating.
  6. XPS is also required for all the samples for correlations.
  7. All the supplementary images should be incorporated into the main text.
  8. The mechanism of antibacterial activity must be presented with detailed explanations.
  9. SEM imaging is required for an antibacterial effect on the Ag coated samples, with comparative to other samples.

Reviewer 2 Report

Dear Authors,

The reviewed manuscript: Molecular Level Polydopamine Thin Film Growth on Mica Surface via In-situ Polymerization of Dopamine, has been structured and written very well by following the main scientific work steps. However, some points should be addressed before publication.

-      AFM operated in AC mode; please add a short sentence describing the mode (contact/tapping etc.) AFM topographical image was used to determined RMS, please add software name and what kind of correction was applied?

-      Is the wettability of the surface was characterized by the static sessile drop method? Also, a short description of ASTM D5946 is needed ( test methods, calculation, type of polar/non-polar liquid). After what the deposition time on the surface, the images of the drop geometry were recorded.

-      Figure 1. – The representative cross-sections line are presented for each topography image.

-      Could the authors add a table with contact angle and RMS values, and it'll improve readability. Could you determine the statistically significantly different groups?

 determine differences between the experimental groups.

-      Unify the surface energy unit - Figure 4.  The whiskers can represent several possible alternative values, add descriptions.

Reviewer 3 Report

This article describes a systematic study of film growth of a PDA layer on mica surface with AFM, contact angle analysis, and XPS. PDA film are very useful for various application and it is worth to mention the film growth of PDA on the surface. As the author describes it needs more efforts to elucidate the PDA structure, it is desirable to add the PDA chemical structures (Polymer) in SI at least. I recommend publication of this paper in Materials after proof reading and revised.

1) At line 35 in page 1, "a surgical baled" would be "a surgical blade".

Round 2

Reviewer 1 Report

The authors have answered all the queries very well, now the paper can be accepted for publication in its present form.